# Association of Positive Bacterial Cultures Obtained from the Throat, Anus, Ear, Bronchi and Blood in Very-Low-Birth-Weight Premature Infants with Severe Retinopathy of Prematurity—Own Observations

**DOI:** 10.3390/jcm12196374

**Published:** 2023-10-05

**Authors:** Monika Modrzejewska, Wiktoria Bosy-Gąsior, Wilhelm Grzesiak

**Affiliations:** 1Scientific Association of Students, II Department of Ophthalmology, Pomeranian Medical University, Powstańców Wielkopolskich 72, 70-111 Szczecin, Poland; wiktoriazbosy@gmail.com; 2Faculty of Biotechnology and Animal Hysbandry, West-Pomeranian Technological University, Al. Piastów 48, 70-311 Szczecin, Poland; wilhelm.grzesiak@zut.edu.pl

**Keywords:** retinopathy of prematurity, bacterial cultures, fungal cultures, swabs from all natural cavities, inflammation parameters for ROP

## Abstract

Background: The causative factors responsible for the development of Retinopathy of Prematurity (ROP) are still unexplored. Therefore, one of the most important factors can be perinatal inflammation. Methods: This retrospective study included 114 premature infants (228 eyes) meeting a birth criteria of ≤ 32 weeks gestational age (GA) and a birth weight (BW) ≤ 1710. Examined Group (EG) *n* = 51 of BW 852.7 ± 255.7; GA 26.3 ± 2.0 with severe ROP treated by diode laser or anti-VEGF intravitreal injection. Control Group (CG) *n* = 63 of BW 1313.9 ± 284.5; GA 28.8 ± 1.6 without ROP. Microbiological bacterial and fungal cultures of the ear, anus, bronchial throat and blood were taken. Medical data and laboratory tests in correlation to 3 ROP and A-ROP were analysed. Results: Positive bacterial tests dominated in EG, 47% vs. CG, 23%. Significant correlations between positive cultures obtained from natural cavities: anus (*p* < 0.001), throat (*p* = 0.002), as well as from blood (*p* = 0.001) and severe ROP which requires diode laser and anti-VEGF treatment were noted. Significant inflammation markers which correlate with the development of severe ROP are *Klebsiella pneumoniae* (*KP*) (*p* = 0.002) and Coagulase-negative *Staphylococci* (CoNS) (*p* < 0.001). CoNS, *p* < 0.001; *KP*, *p* = 0.002; the remaining *Maltophilia stenotrophomonas* (*MS*); *Staphylococcus aureus* (*SA*), *p* = 0.005; and *Enterobacter cloacae* (*EC*), *p* = 0.02 were the most frequent bacteria in severe ROP. High levels of white blood cells (WBC), C-reactive protein (CRP), lymphocytes (LYM) and low thrombocytes (PLT) correlated sequentially with (Odds Ratio, OR) *CoNS* (2.3); *MS* (5.9); *KP* (3.1); and *all positive cultures* (*APC*) (9.5). An important correlation between the BPD—*EC* (4.3); intrauterine inflammation—*KP* (3.4); PDA—*EC* (3.9); and asphyxia—*CoNS* (3.0) was identified. Conclusions: It cannot be ruled out that positive microbiological results of blood, anal and pharyngeal cultures may become prognostic markers for the early development of ROP, which would enable early initiation of ophthalmological treatment in premature infants from the VLBW group.

## 1. Introduction

Retinopathy of prematurity (ROP) is a retina disease based on morphological and physiological immaturity of preterm infants and correlated with the so-called causative factors responsible for the development of ROP. 

There have been 23 factors described until now, among them not only toxic oxygen radicals but also growth factors (VEGF, IGF-1, BDNF), metaloproteinases (MMP-1, MMP-9), HIF1-α i HIF2-α, EPO, PlGF, bFGF/FGF-2, Ang1, Ang 2, TSH, IL-1β, TNF-α, IL-6, IL-8, RANTES, MCP-1, I-TAC and semaphorins [1]. Alongside these factors, infections accompanied by a weak immune system are key modulators for severe ROP.

The incidence rate of bacterial infections among extreme VLBW prematures is 96%. Most often, these are *Coagulase-negative Staphylococci* (*CoNS*); 75%, fungal; 4% (including *Candida albicans*; 96%) [2] and viral; 12.7% (mostly *rotaviruses*; 23.5%, *Respiratory Syncytial* Virus (RSV); 16.2% and *enteroviruses*; 16.2%) [3], for which the death rate equals 62%, 3% and 12%, respectively [4]. In comparison, the death rate among prematures without infections is 35% [5].

It is significant that the severity of ROP increases with the development of infections in the perinatal period by 44.8% (and 50.77% in Italy [6] and 41.4% in USA) [7], in which the most common factor for premature birth is chorioamnionitis, which is of bacterial etiology (Gram-positive bacteria; *TORCH*), which originates in 50% of prelabour rupture of membranes (PPROM) [8]. Also, current research considers COVID-19 as a causative factor [9]. 

Dysbiosis within the microbiome in premature infants may also be evidence of an immune imbalance, as reported by new reports in the scientific literature [10,11,12,13]. The correlation between the microbiome and ROP has been demonstrated, among others, by Skonda et al. [10]. The increase in factors such as IL-6, IL-1β, TNF-α and VEGF-A observed in gut dysbiosis now also correlates with pathological angiogenesis in the retina [11]. Similarly, the concentration of the IGF-1 factor (which was used in the WINROP algorithm [14]) is also described as key in intestinal microbiota disorders, which may constitute evidence of the connection between the gut–retina axis and the mechanism of ROP development [12].

This research aims to clarify the relation between positive bacterial and fungal cultures obtained from four natural body cavities (throat, ear, anus and bronchi) in infants and from blood using inflammatory parameters in lab tests and severe ROP treated with diode lasers and anti-VEGF intravitreal injections.

## 2. Methods

### 2.1. Patients

The study cohort established for this retrospective case–control study was obtained from medical records collected by the Ophthalmology Prematurity Clinic of the Pomeranian Medical University in Poland between June 2017 and October 2021 and includes 114 prematurely born neonates (*n* = 114; e = 228 eyes) who met all criteria set for the study: neonatal, ophthalmological, laboratory and microbiological.

The ophthalmological assessment of the retina during further individualized screenings was performed with the use of RetCam3.

The clinical state of ROP was evaluated on the basis of the previous International Classification of ROP [15], applicable at the time of the clinical trial (1ROP—5ROP and aggressive ROP (A-ROP)), in the presence or absence of the disease, “plus”.

### 2.2. Study Cohorts

The patients were divided into two groups: Examined Group (EG) (*n* = 51) and Control Group (CG) (*n* = 63) according to clinical stages of ROP, but for statistical analyses were unified according to birth age and weight. Eligibility criteria included birth weight (BW) ≤ 1710 g and gestational age (GA) ≤ 32; diagnosis of 3ROP, 5ROP and A-ROP which were treated with diode laser or anti-VEGF intravitreal injections (Ranibizumab) or combined laser and anti-VEGF. Additionally, 6 subgroups according to birthweight were distinguished so that pathogens characteristic for each subgroup could be assessed.

### 2.3. Data from Neonatal Hospital Wards

The medical data comprised of positive microbial smear results from 4 natural body cavities (throat, bronchi, ear, anus) and blood obtained during the hospitalization of premature babies directly after birth and before qualifying for ROP treatment (up to 72 h after diagnosis) with a mean GA 34 ± 1. Moreover, medical data from the pre- and postnatal period as well as lab data were analysed in the examined group of children.

### 2.4. Microbiological Swabs and Blood 

Swabs from four body orifices were taken with brush spatulas which were secured in containers with transport medium and transported as quickly as possible to the microbiology laboratory at 20–24 °C. This is a routine procedure performed in the neonatal intensive care unit to detect alarm pathogens.

In the next step, the collected biological material was transferred to a suitable culture medium for microbial growth (Petri dish). The material was placed on the medium for bacteria (sheep blood agar) and for fungi (Sabouraud’s medium). Successively, the dishes and trays with smear cultures were closed in incubators at 37 °C for bacteria and 30 °C for fungi for 48–72 h. 

The same procedure was followed with the collected blood, which from the moment of collection was placed in a special container with liquid medium broth (containing special resins that inhibit the effect of antibiotics used earlier). The tubes were placed in the apparatus at 37 °C for bacteria and 30 °C for fungi for 48–72 h. Bacterial colonies were then observed under a microscope and the result for alarm microorganisms was sent to the neonatology department.

#### Statistical Analysis

Comparisons between the EG and CG parameters were conducted with the use of Student’s *t*-test and, in the case of exceptions from normal distribution (parameters and/or unequal variables in researched groups), a nonparametric Mann–Whitney rank-sum test was used. In order to reduce differences between the parameters GA, BW and Apgar score, the Fisher test was applied. In order to compare nominal variables in demographic data and lab parameters, Yates’s correction for continuity was applied. The test was also used for checking the presence of bacteria in general and in detailed cultures. The test was used to confirm the incidence of positive bacterial cultures in the 4 physiological cavities and blood. The type of applied test is highlighted in the tables. The assumed significance level was α = 0.05 and the statistically significant differences were for *p* ≤ 0.05.

The odds ratio (OR) was calculated as an odds ratio of two comparable groups where OR = 1 means risk equivalence of the compared groups. In OR > 1, the odds of the occurrence of an outcome in group A are higher than in group B and vice versa in OR < 1. The results including OR ≥ 1.5 were plotted on a log scale (Cl 95%). The OR was used to verify the correlation of associated comorbidities and lab parameters of selected bacteria. Calculations were carried out in Statistica13 software.

## 3. Results

### 3.1. Demographic Data 

There were two groups of prematures selected: *n* = 51 babies treated with diode laser due to severe ROP. In EG number of eyes (e) 3ROP, e = 73; 5ROP, e = 2; A-ROP, e = 26 included: e = 58, eyes treated with the diode laser; e = 8, anti-VEGF intravitreal injections; e = 36, after combined therapy of diode laser and anti-VEGF. Next, *n* = 63 infants constituting CG; without ROP: e = 100; ROP 1, ROP 2 after spontaneous regression: e = 26.

The researched groups showed statistical differences between EG and CG in birth weight (BW) (*p* < 0.001), GA (*p* < 0.001) and average Apgar score (*p* < 0.001).

In order to reduce the differences between the groups, body weight was integrated in the calculations and differences were established in GA (*p* = 0.03), but not in the analysis of the average Apgar score (*p* = 0.93).

In both groups, females (*p* = 0.66), single pregnancy (*p* = 0.78) and caesarean delivery (*p* = 0.01 for CG) prevailed (Table 1; part I).

The description of the patients in reference to body weight is presented in groups A-F (Table 1; part II).

Premature babies are compared according to weight (B vs. D *p* < 0.001; C vs. E *p* = 0.05), GA (B vs. D *p* < 0.001; C vs. E *p* = 0.002) and average result of Apgar score (B vs. D *p* = 0.01; C vs. E *p* = 0.33).

The following risk factors have been noted in EG: anaemia (EG 96.1% (49 of 51) vs. CG 65.1% (41 of 63); *p* < 0.001) and blood transfusions (EG 96.1% (49 of 51) vs. CG 66.7% (42 of 63); *p* < 0.001).

There have been no statistical differences in RBC measurement (below the lower limit) in the first 24 h (EG 33.3% (17 of 51) vs. CG 28.6% (18 of 63), *p* = 0.73); however, the average number of red blood cells during the whole stay at the neonatal ward was significantly lower in EG in comparison to CG (EG 2,7 vs. CG 2,9 [T/l]; *p* = 0.003). The lab blood count of selected parameters during the first 24 h of the infant’s life and the ones obtained prior to ophthalmological treatment (tested in the second week: EG mean GA 28 ± 1, CG 30 ± 1) included White Blood Cells (WBC), neutrophils (Neu), lymphocytes (LYM), platelet count (PLT), C-reactive protein (CRP) and procalcitonin (PRC). The lab results obtained during the first 24 h reveal statistically significant differences between the levels of WBC, PLT and the measurements taken prior to the treatment: CRP and LYM (Figure 1, Appendix A). Laboratory test results indicating reduced parameters were based on the norms described in the literature for the exact days of a premature infant’s life [16] (Figure 1, Appendix A, Table 2).

### 3.2. Microbiology

In the analysed bacterial cultures, there were 25 types of bacteria identified (*Escherichia coli*, *Acinetobacter baumanii*, *Klebsiella pneumoniae*, *Enterococcus faecialis*, *Enterococcus faecium*, *Staphylococcus epidermidis*, *Klebsiella oxytoca*, *Staphylococcus haemolyticus*, *Pseudomonas aeruginosa*, *Citrobacter freundii*, *Klebsiella aerogens*, *Morganiella morgani*, *Enterobacter cloacae*, *Serratia marcescens*, *Staphylococcus aureus*, *Staphylococcus capitis*, *Maltophilia stenotrophomonas*, *Enterobacter asburiae*, *Staphylococcus hominis*, *Enterobacter kobei*, *Ureaplasma*, *Citrobacter farmerii*, *Citrobacter braaki*, *Acinetobacter Iwoffii*, *Citrobacter murliniae*), of which 11 were selected as the most frequent. The bacteria were examined in order to determine the exact bacteria family and the quantity of positive cultures (m) obtained from the four physiological cavities and blood of each child.

Out of all the cultures, a substantially larger number of pathogens was detected in EG (47.1% (120 of 255) than in CG (22.5% (71 of 315). 

The study demonstrates that the family of *Coagulase-negative Staphylococci* (*CoNS*) has the greatest influence on the development of severe ROP. Another of equal frequency appears to be *Klebsiella pneumoniae* (*KP*). The results indicate that other bacteria may be significantly important in the development of severe ROP when positive cultures are simultaneously obtained with higher multiplicity from the body cavities (Table 2).

The total amount of positive cultures from each of the physiological cavities separately: the throat, anus and blood, for each premature baby indicated statistically significant differences between the groups (throat: EG 47.1% [24 of 51] vs. CG 20.6% [13 of 63], *p* = 0.002; anus EG 72.5% [36 of 51] vs. CG 36.5% [23 of 63], *p* < 0.001; blood EG 43.1% [22 of 51] vs. CG 15.9% [10 of 63], *p* = 0.001) (Figure 2). 

Statistical significance was also calculated for positive bronchi (*p* = 0.02) and ear (*p* = 0.43) culture. Nevertheless, for further calculations, three of the most frequent physiological cavities were selected from the above. 

The outcomes suggest that bacteria of the greatest significance in the development of ROP in positive throat, anus and blood cultures are: *KP*, *Enterobacter cloacae* (*EC*), *Staphylococcus aureus* (*SA*) and bacteria of the *CoNS* family. The most significant for ROP appeared to be sepsis of *CoNS* etiology: *Staphyloccus epidermidis* (EG 16.6% [10 of 51] vs. CG 6.4% [4 z 63], *p* = 0.03) and *Staphylococcus haemolyticus* (EG 13.7% [7 of 51] vs. CG 0% *p* < 0.001) (Figure 2).

The percentage of bacteria in blood cultures was presented in the examined groups (Figure 2): all neonates (EG 43.1% [22 of 51] vs. CG 15.9% [10 of 63]), neonates with positive cultures (EG 46.8% [22 of 47] vs. CG 34.5% [10 of 29]) and neonates fulfilling sepsis criteria (EG 59.1% [13 of 22] vs. CG 60% [6 of 10]). 

Bacteria which manifested the greatest statistical significance in prematures of 1500–1000 g weight (Table 1; part II) include: *KP* (in general bacterial cultures: EG 43.8% [7 of 16] vs. CG 6.3% [2 of 32], *p* = 0.01; in detailed bacterial cultures: EG 9% [7 of 80] vs. CG 1% [2 of 160], *p* = 0.01) and *CoNS* (in general bacterial cultures: EG 62.5% [10 of 16] vs. CG 12.5% [4 of 32], *p* = 0.001; in detailed bacterial cultures: EG 21% [17 of 80] vs. CG 6% [9 of 160], *p* < 0.001). No statistically significant pathogens were detected in the remaining weight subgroups (*p* > 0.05). 

Furthermore, microbiological examination showed a fungal infection with *Candida albicans*, which was statistically significant between groups (EG 9,8% [5 of 51] vs. CG 1,6% [1 of 63]; *p* = 0.04). Three infants had *Candida* confirmed consecutively prior to each treatment procedure as fungemia in EG.

### 3.3. Odds Ratio

The occurrence of significant pathogens in the development of ROP (Table 3) was compared with comorbidities (*p* ≤ 0.05) and inflammatory parameters (*p* ≤ 0.05) in the EG and CG groups, calculating the odds ratio (Table 3; part I, II).

## 4. Discussion

It has been already proven that in a widely developed pathogenetic mechanism [1], an important role is played by i.a.: cytokines, chemokines, factors inducing hypoxia, hormones and growth factors, as well as infection cells such as leukocytes, monocytes and microfags/microglia. All of these taking part in the angiogenesis control determine their role in the development of retina blood vessels in the progression of ROP [1,17,18,19]. 

The authors raise an interesting question regarding the analysis of the correlation between the type of pathogen and the development of severe ROP in VLBW prematures.

In this retrospective case–control study, the authors explore the relation between inflammation (assessed by the level of infection markers) and type of bacteria (microbial cultures) in relation to ROP. 

The literature confirm that postnatal multi-organ disfunctions such as respiratory distress syndrome (RDS), bronchopulmonary dysplasia (BPD) or necrotizing enterocolitis (NEC) often occur among VLBW [20,21] neonates and, according to Wu et al. [22], strongly correlate with severe ROP. Studies carried out by Chiang et al. [23] and Lundgren et al. [24] reveal that late-onset sepsis (LOS), frequent in VLBW prematures born prior to 32 GA [2], is a confirmed risk factor in the pathogenesis of severe ROP and A-ROP. The above is supported by the findings in this study, which confirm that sepsis incidence was substantially more frequent among prematures with severe ROP than those without ROP: 43.1% and 15.9%, respectively (*p* < 0.001) (Figure 2).

In both groups, positive blood culture was detected in ca. 60% with positive bacterial cultures from the anus, throat or both, and with a manifestation >1 of preneonatal disease of inflammation etiology. The most frequent etiology factor of positive blood culture were bacteria of the *CoNS* type (35.3%; *p* < 0.001) (Figure 2), which was also confirmed in studies from Weintraub et al. [25] and British neonIN (Neonatal Infection Surveillance Network) [2], in which *CoNS* constituted the most frequent factor for LOS. The authors additionally emphasize the coexistence of neurological seizures with sepsis connected with *CoNS* (OR = 3.0, 95% Cl: 0.1, 73.6) and the development of ROP, as opposed to results by Cantey et al. [26] and Ohlin at al. [27] in which the presence of *CoNS* was not associated with ROP.

There is an interesting observation from Stewart et al. [28] indicating a correlation between the presence of *Enterobacteriacae* in the stool cultures of 1-month prematures and the occurrence of LOS. The stool smear results were different from the pathogens present in the blood of the same group of infants and constituted a separate bacterial panel. The most frequent pathogens detected in sepsis were: *Klebsiella spp.*, *Enterococcus* spp., *Staphylococcus* spp. And *Escherichia* spp., along with a lower amount of commensal bacteria of the *Bifidobacterium* type. The research conclusions from the above-mentioned authors are consistent with the results of this study and they demonstrate that the type of bacteria from microbial cultures are of significant importance in the development of severe ROP—and so, for *Klebsiella* spp. *p* < 0.001, for *Enterobacter cloacae p* = 0.02 (Table 2). The same bacteria showed statistically significant correlations not only with severe ROP but also for *KP* with positive throat smear, *p* = 0.02 and, consecutively, for with anus smear, *p* = 0.001, and for *EC* with positive throat smear, *p* = 0.04 (Figure 2). This observation indicates that the presence of multi-organ infection coincides with ROP.

One of the common diseases coinciding with VLBW infants is BPD. Its presence correlates not only with severe ROP but also with bacterial pathogens found in this group of neonates such as *EC* (OR = 4.3, 95% Cl: 0.9, 21.4), *SA* (OR = 2.5, 95% Cl: 0.3, 22.9), *CoNS* (OR = 1.6, 95% Cl: 0.5, 3.6) and KP (OR = 1.5, 95% Cl: 0.4, 6.3) (Table 3; part I).

There are reports describing similar mechanisms of pathogenetic development of ROP and BPD, in which a major role is played by inflammatory factors monocyte chemotactic protein 1 (MCP-1/CCL2), macrophage inflammatory protein 1 alpha (MIP-1α/CCL3), interleukin 6 (IL-6), macrophage inhibitory factor and angiogenne (fibroblast growth factor 1 (FGF-1) and the platelet-derived growth factor (PDGF) [29]. Our observation is also confirmed by the research of Podraza et al. [30] in which BPD correlated with severe ROP, *p* = 0.006, and Singh et al. [31], *p* < 0.001. Researchers Bae et al. [32] confirmed the results of the above studies and additionally emphasized the significance of neurological factors such as IVH in the development of severe ROP. Many authors consider this factor as highly important and postulate IVH prevention in order to inhibit the development of ROP [33,34]. The results of this study are similar to the above and they demonstrate the correlation not only with severe ROP but also with the presence of *EC* (OR = 1.1, 95% Cl: 0.38, 3.24) and *CoNS* (OR = 1.7, 95% Cl: 0.1, 22.5) (Table 3). Research conducted by Hand et al. [35] remains controversial as it does not recognize any correlation between IVH and the development of severe ROP. Another important disease which correlates with ROP in VLBW prematures is NEC, which is, according to Soraisham et al. [36], linked with vision impairment and neurodevelopment disorder. It is worth adding that recent publications show a connection between inflammation and the regulation of intraneuronal signals (Semaphorins 3A class, Sema3A) which exist in activated microglia cells present in the hypoxic retina and which damage the endothelium of the retina blood vessels, contributing this way to the pathogenesis of ROP [1]. In the Masi et al. [37] study, the type of pathogens in NEC confirmed a mechanism etiologically compatible with the type of bacteria obtained in Stewart et al.’s [28] study. However, unlike the former, the important correlation was linked to *EC*, which was detected in larger amounts in stool cultures prior to symptoms, suggesting the development of a full-blown NEC [37]. The results in the presented study confirm a statistically significant relation between NEC and severe ROP (*p* = 0.004) (Table 3). Our observations indicate that EC was detected in throat and anus cultures, but the statistical analysis does not demonstrate a direct relation with the development of NEC and severe ROP (OR < 1) (Table 3). Moreover, the authors emphasize that a highly substantial pathogen for severe ROP is *KP*, which appeared to be a statistically frequent bacteria detected in throat and anus cultures obtained from prematures with severe ROP, at 8.6% and 22.5%, respectively (Figure 2). Furthermore, an increase in the frequency of *KP* in anus and throat cultures was observed in babies with severe ROP who had previously experienced intrauterine infection (OR = 3.4, 95% Cl: 0.4, 33.8) (Table 3).

In search engines such as PubMed, GoogleScholar and UpToDate, in the years 2011–2021 there is only one article by Weintraub et al. [25] on the relation between positive blood culture results with a type of bacteria and the development of severe ROP, in which a correlation was detected with bacteria families of *CoNS*, *Klebsiella*, *Enterobacter* and *Serratia*. A similar dependence was identified by Menke et al. [38] in a case report, in which *KP* from a nasopharynx smear was associated with ROP 3, treated with anti-VEGF intravitreal injections. The association of the development of ROP with the disfunction of the microbiotic composition in the gastrointestinal tract (mainly *Enterobacteriaceae*) mentioned i.a. in the study of Skondra et al. [10].

Similarly to as described above, the fungal pathogen, which originates in the form of *Candida albicans* in 96% of cases [39], plays an important role in the development of ROP (incidence rate: 4%). Research conducted by Mittal et al. [40] and Kremer et al. [41] confirmed the correlation between fungemia and the development of ROP. An important argument which supports the above theory is the presence of *Candida albicans*, which correlates with severe ROP in the cohort examined in this study (*p* = 0.04) (results; microbiology). It is worth highlighting the relation between *Candida* infection and severe ROP in babies treated with diode lasers and supported with anti-VEGF intravitreal therapy. A different result has been observed in Karlowicz et al.’s [42] study.

The association of ROP with inflammation manifested by the increase of CRP is confirmed by Ikeda et al. [43]. The increase of PRC and IL-6 can be yet another predictor for ROP, which Kurul et al. [44] mention as more specific inflammation markers in ROP and are also confirmed by Andreola et al. [45]. This study confirms the importance of the CRP level prior to therapy implementation (*p* = 0.001) and shows how statistically significant it is for inflammation in ROP (*p* > 0.05), which is also confirmed by Singh et al. [46].

Moreover, in correlation with positive bacterial cultures, this parameter turned out to be a key indicator of infection in the child (Appendix A, Figure 1). This study identified a substantially higher frequency of bacteria occurrence with a high level of CPR prior to ROP treatment (Table 3). There is also research which confirms the correlation between ROP and the PRC increase without considering other inflammation parameters [47].

Another inflammation parameter mentioned in the literature is the neutrophile–lymphocytes ratio (NLR), which is applied in the diagnosis of inflammation in diseases in which “physiological stress” [48,49,50,51] is a pathophysiological factor. However, further studies by Kurtul et al. [52] and Ozturk et al. [53], who tested this parameter for the development of ROP, have not confirmed this observation. It is worth noting that in this study the NLR parameter did not show any important correlation with severe ROP (*p* > 0.05) (Appendix A, Figure 1) either. Furthermore, the research confirmed significantly higher lymphocyte levels in the second week of the child’s life and also revealed a more frequent correlation between bacteria and *Candida albicans* and lymphocytosis in babies with severe ROP. (Appendix A, Table 3). In line with other research, this study identified thrombocytopenia prior to ophthalmological treatment in the examined cohort of preneonates. There is much research on the importance of low levels of blood cells and their predicative role for the development of ROP, which confirms the significance of this assumption [54,55].

This study demonstrates that the mean cell count in the cohort with ROP also turned out to be significantly lower than in the control cohort in the first 24 h (*p* = 0.004) (Figure 1). Moreover, thrombocytopenia occurs significantly more often in preneonates with positive bacterial cultures and who later developed severe ROP (Table 3).

## 5. Conclusions

According to the authors, positive bacterial culture infections are connected with severe ROP. This observation is confirmed by significant correlations between positive cultures obtained from natural cavities, the anus (*p* < 0.001) and throat (*p* = 0.002), as well as from blood (*p* = 0.001), and severe ROP which requires diode laser and anti-VEGF treatment. Significant inflammation markers which correlate with the development of severe ROP are *KP* (*p* = 0.002) and *CoNS* (*p* < 0.001).

There was a statistically significant relation confirmed between thrombocytopenia and increased WBC, CRP and lymphocytes (during neonate’s first hours of life) connecting with an ROP diagnosis. Positive bacterial cultures (such as *CoNS*, *KP*, *SA*, *EC*, *MS*) as well as positive fungal outcomes (*Candida albicans*) appeared more often in preterm infants with severe ROP. 

High levels of white blood cells (WBC), C-reactive protein (CRP), lymphocytes (LYM) and low thrombocytes (PLT) correlated sequentially with (OR): *CoNS* (2.3); *MS* (5.9); *KP* (3.1) and *APC* (9.5). An important correlation between the BPD—*EC* (4.3); intrauterine inflammation—*KP* (3.4); PDA—*EC* (3.9); and asphyxia—*CoNS* (3.0) was identified.

It is possible that positive results of microbial cultures of blood as well as anus and throat cultures may jointly become prognostic markers for the early development of ROP, which will enable early implementation of ophthalmological treatment among VLBW preterm newborns.

## 6. Limitations

The number of examined patients in this study is a limitation. However, we continue to gather data on neonates with severe ROP in order to create a bigger study group and present new data in the future.

## Figures and Tables

**Figure 1 jcm-12-06374-f001:**
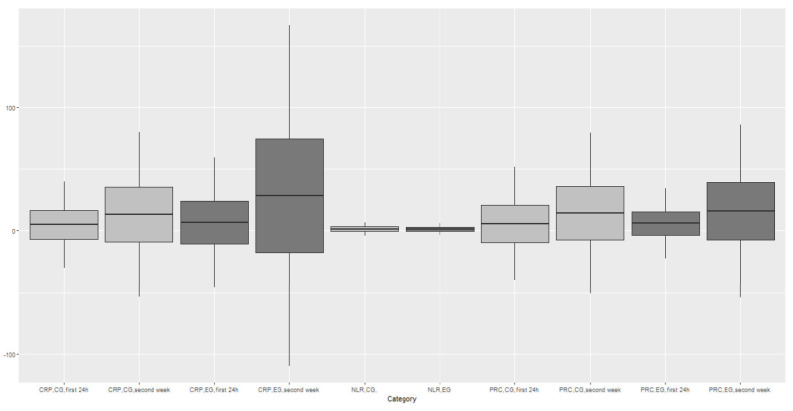
Results of laboratory and morphology parameters of the examined groups of premature babies.

**Figure 2 jcm-12-06374-f002:**
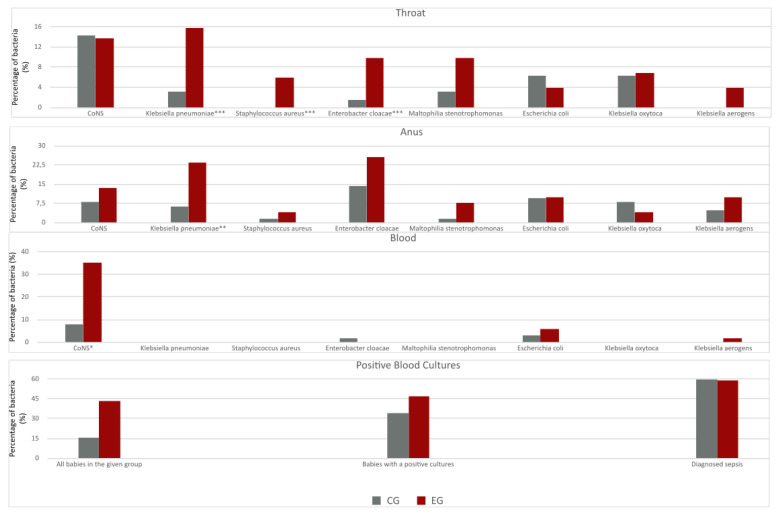
Graphs indicating the presence of positive bacterial cultures/%/obtained from the throat, anus and blood cultures in the group of premature babies with severe ROP compared to the group of premature babies without ROP. Sepsis was diagnosed on the basis of the following criteria met: (1) positive blood cultures; (2) positive anus and/or throat cultures; (3) manifestation >1 of pre-neonatal disease of inflammation etiology. Calculations performed with chi2 and chi2 Yates’s correction. Differences statistically significant: * *p* value < 0.001, ** *p* value 0.001–0.01, *** *p* value 0.01–0.05.

**Table 1 jcm-12-06374-t001:** Demographic characteristics of Baseline Values (part I, subgroups (A-F) selected according to birth weight (part II).

Part I	CG *n* = 63 ^a^			EG *n* = 51 ^b^			*p* Value
**BW**, g							
Mean ± SD	1313.9 ± 284.5			852.7 ± 255.7			<0.001
Median (IQR)	1390.0 (650.0–1710.0)			850.0 (400.0–1500.0)			
**GA**, weeks							
Mean ± SD	28.8 ± 1.6			26.3 ± 2.0			<0.001
Median (IQR)	30.0 (26.0–32.0)			27.0 (22.0–29.0)			
Sex							
Male	*n* = 31 (49%)			*n* = 23 (45%)			
Female	*n* = 32 (51%)			*n* = 28 (55%)			
Average Apgar Score ^c^	6.3 ± 1.7			5.1 ± 1.8			<0.001
Pregnancy							
Single	*n* = 48 (76%)			*n* = 40 (78%)			
Multiple	*n* = 15 (24%)			*n* = 11 (22%)			
Birth							
Caesarean section	*n* = 59 (94%)			*n* = 39 (78%)			
Vaginal birth	*n* = 4 (6%)			*n* = 12 (24%)			
Ventilation ^d^	*n* = 33 (52%)			*n* = 48 (94%)			<0.001
Mechanical, (mean SD), days ^e^	6.5 ± 11.8			22.6 ± 17.2			<0.001
Supported, (mean SD), days ^f^	6.7 ± 12.2			21.6 ± 17.1			<0.001
Part II	**Subgroups based on birth weight**						
	CG			EG			*p* value
Subgroups	Group A >1501 g *n* = 18|m = 90	Group B 1500–1000 g *n* = 32|m = 160	Group C 980–650 g *n* = 13|m = 65	Group D 1500–1000 g *n* = 16|m = 80	Group E 970–650 g *n* = 23|m = 115	Group F <650 g *n* = 12|m = 60	
BW, g							
Mean ± SD	1619.4 ± 50.8	1325.2 ± 130.1	863.1 ± 101.3	1153.4 ± 178.0	798.5 ± 88.1	555.8 ± 67.1	<0.001 ^g^ 0.05 ^h^
Median (IQR)	1615 (1550–1710)	1345 (1000–1500)	900 (650–980)	1082.5 (1000–1500)	800 (650–970)	575 (400–630)	
GA, weeks							
Mean ± SD	31.1 ± 0.8	30 ± 1	27.6 ± 1.3	28 ± 1.1	25.9 ± 1.5	24.9 ± 2.2	<0.001 ^g^ 0.002 ^h^
Median (IQR)	31 (30–32)	30 (28–32)	27 (26–30)	28 (26–29.5)	25.5 (23.5–29)	24 (22–28.5)	
Average Apgar Score ^c^	7.4 ± 1.6	6.7 ± 1.3	5.9 ± 2.2	5.5 ± 1.8	5.3 ± 1.7	4.7 ± 1.6	0.01 ^g^ 0.33 ^h^
Sex							
Male	*n* = 9 (50%)	*n* = 14 (44%)	*n* = 7 (54%)	*n* = 7 (44%)	*n* = 12 (52%)	*n* = 4 (33%)	
Female	*n* = 9 (50%)	*n* = 18 (56%)	*n* = 6 (49%)	*n* = (56%)	*n* = 11 (48%)	*n* = 8 (67%)	

^a^ Number of premature babies in CG. ^b^ Number of premature babies in EG. ^c^ Average in minute 1, 3 and 5 of life. ^d^ Total number of days of mechanical and supported ventilation. ^e^ Applied respirators for mechanical ventilation: SNIPPV, SIMV, VIVE, VG, HFNC, HFO. ^f^ Applied equipment for supported ventilation: nCPAP, Neopuff, Infant-Flow; +AP, Vapotherm, Optiflow. ^g^ Group D vs. group B. ^h^ Group E vs. group C. *p* value checked by U Mann–Whitney test and χ2 Yates’s correction for continuity; BW- birth weight; GA-gestational age; Mean-an average numerical value calculated mathematically.

**Table 2 jcm-12-06374-t002:** Comparison of positive bacterial cultures between groups CG vs. EG and their correlation with the development of severe ROP.

	Bacterial Cultures from 4 Cavities and from Blood *n* = 114							Detailed Bacterial Cultures ^2^				General Bacterial Cultures ^3^		
	Throat	Anus	Bronchi	Ear	Blood	^1^ *p* value	CG m = 315 ^a^		EG m = 255 ^b^	^2^ *p* value	CG *n* = 63 ^c^		EG *n* = 51 ^d^	^3^ *p* value
*Coagulase-negative Staphylococci* ^e^	*n* = 16 (14%)	*n* = 12 (11%)	*n* = 3 (3%)	*n* = 1 (1%)	*n* = 23 (20%)	<0.001	m = 20 (6%)		m = 35 (14%)	0.003	*n* = 12 (19%)		*n* = 26 (51%)	<0.001
*Staphylococcus epidermidis*	*n* = 3 (3%)	*n* = 4 (4%)	*n* = 2 (2)	*n* = 0 (0%)	*n* = 14 (12%)	<0.001	m = 8 (1%)		m = 15 (3%)	0.04	*n* = 7 (11%)		*n* = 13 (26%)	0.08
*Staphylococcus haemolyticus*	*n* = 5 (4%)	*n* = 1 (1%)	*n* = 1 (1%)	*n* = 0 (0%)	*n* = 7 (6%)	0.005	m = 6 (1%)		m = 8 (1%)	0.35	*n* = 4 (6%)		*n* = 8 (16%)	0.19
*Klebsiella pneumoniae*	*n* = 10 (9%)	*n* = 17 (15%)	*n* = 1 (1%)	*n* = 0 (0%)	*n* = 0 (0%)	<0.001	m = 6 (2%)		m = 22 (9%)	<0.001	*n* = 4 (6%)		*n* = 14 (28%)	0.002
*Enterobacter cloacae*	*n* = 6 (5%)	*n* = 22 (19%)	*n* = 2 (2%)	*n* = 0 (0%)	*n* = 1 (1%)	<0.001	m = 11 (4%)		m = 20 (8%)	0.02	*n* = 10 (16%)		*n* = 15 (29%)	0.08
*Eschericha coli*	*n* = 6 (5%)	*n* = 11 (10%)	*n* = 2 (2%)	*n* = 5 (4%)	*n* = 5 (4%)	>0.99	m = 18 (6%)		m = 11 (4%)	0.45	*n* = 8 (13%)		*n* = 8 (16%)	0.85
*Maltophilia stenotrophomonas*	*n* = 7 (6%)	*n* = 5 (4%)	*n* = 2 (2%)	*n* = 0 (0%)	*n* = 0 (0%)	0.001	m = 3 (1%)		m = 11 (4%)	0.005	*n* = 3 (5%)		*n* = 7 (14%)	0.09
*Staphylococcus aureus*	*n* = 3 (3%)	*n* = 3 (3%)	*n* = 3 (3%)	*n* = 0 (0%)	*n* = 0 (0%)	0.05	m = 1 (0.3%)		m = 8 (3%)	0.005	*n* = 1 (2%)		*n* = 5 (10%)	0.05
*Klebsiella oxytoca*	*n* = 6 (5%)	*n* = 7 (6%)	*n* = 1 (1%)	*n* = 0 (0%)	*n* = 0 (0%)	<0.001	m = 9 (3%)		m = 5 (2%)	0.49	*n* = 6 (10%)		*n* = 2 (4%)	0.23
*Klebsiella aerogens*	*n* = 2 (2%)	*n* = 8 (7%)	*n* = 0 (0%)	*n* = 0 (0%)	*n* = 1 (1%)	0.001	m = 3 (1%)		m = 8 (3%)	0.06	*n* = 3 (5%)		*n* = 6 (12%)	0.17

^a^ Number of cultures in CG; ^b^ Number of cultures in EG; ^c^ Number of preneonates in CG; ^d^ Number of preneonates in EG; ^e^ CoNS group includes among others: Staphylococcus epidermidis, Staphylococcus haemolyticus. ^1^ The total number of cultures from each of the four physiological cavities and blood, as a total derived from both CG and EG (*n* = 114). ^2^ Detailed cultures indicate the total number of bacterial cultures collected from four physiological cavities (throat, anus, bronchi, ear) as well as blood, depending on the type of bacteria in each child (m = 100%); *p* value was calculated for the frequency of cultures taken from the physiological cavities and blood (the statistical differences shown between the number of cultures from each source). ^3^ General bacterial cultures interpreted as positive cultures indicating a given bacteria in each neonate separately (*n* = 100%); *p* value was calculated for the frequency of total positive cultures obtained from each cavity and positive blood cultures between CG vs. EG cohorts. Calculations have been carried out with chi2 test with Yates’s correction; *p* value, statistically significant parameter.

**Table 3 jcm-12-06374-t003:** Odds ratio for pathogen occurrence in premature diseases and comorbidities in the group with severe ROP.

			CG *n* = 63	EG *n* = 51	Pathogen Significant in Disease a	OR b	95%Cl
Part I premature diseases/comorbidities ^c^		BPD	*n* = 13 of 20 (65%)	*n* = 40 of 43 (93%)	APC	7.2	1.6, 31.9
			*n* = 2 of 20 (10%)	*n* = 14 of 43 (33%)	EC	4.3	0.9, 21.4
			*n* = 1 of 20 (5%)	*n* = 5 of 43 (12%)	SA	2.5	0.3, 22.9
			*n* = 8 of 20 (32%)	*n* = 22 of 43 (51%)	CoNS	1.6	0.5, 4.6
			*n* = 3 of 20 (15%)	*n* = 9 of 43 (21%)	KP	1.5	0.4, 6.3
		Intrauterine infections	*n* = 7 of 9 (78%)	*n* = 19 of 20 (95%)	APC	5.4	0.4, 69.7
			*n* = 1 of 9 (11%)	*n* = 6 of 20 (30%)	KP	3.4	0.4, 33.8
		Asphyxia	*n* = 2 of 5 (40%)	*n* = 13 of 14 (93%)	APC	19.5	1.3, 292.8
			*n* = 1 of 5 (20%)	*n* = 6 of 14 (43%)	CoNS	3.0	0.3, 34.2
		IVH III/IV grade	*n* = 2 of 4 (50%)	*n* = 10 of 11 (91%)	APC	10.0	0.58, 171.2
			*n* = 1 of 4 (25%)	*n* = 4 of 11 (36%)	CoNS	1.7	0.1, 22.5
		PDA	*n* = 4 of 10 (40%)	*n* = 18 of 20 (90%)	APC	13.5	2.0, 93.3
			*n* = 1 of 10 (10%)	*n* = 6 of 20 (30%)	EC	3.9	0.4, 37.9
			*n* = 4 of 10 (40%)	*n* = 12 of 20 (60%)	CoNS	2.3	0.5, 10.6
			*n* = 1 of 10 (10%)	*n* = 4 of 20 (20%)	SA	2.3	0.2, 23.3
		Seizures	*n* = 1 of 2 (50%)	*n* = 7 of 8 (88%)	APC	7.0	0.2, 226
			*n* = 1 of 2 (50%)	*n* = 6 of 8 (75%)	CoNS	3.0	0.1, 73.6
		NEC	*n* = 1 of 2 (50%)	*n* = 9 of 10 (90%)	APC	9.0	0.3, 285.5
		Hernia ^f^	*n* = 2 of 3 (67%)	*n* = 10 of 11 (91%)	APC	5.0	0.21, 117.9
Part II Laboratory parameters ^d^	First 24 h of neonate’s life	WBC	*n* = 5 of 7 (71%)	*n* = 25 of 27 (93%)	APC	5.0	0.6, 44.3
			*n* = 2 of 7 (29%)	*n* = 13 of 27 (48%)	CoNS	2.3	0.4, 14.1
			*n* = 2 of 7 (29%)	*n* = 10 of 27 (37%)	KP	1.5	0.2, 9.0
		Thrombocytopenia	*n* = 6 of 9 (48%)	*n* = 19 of 20 (95%)	APC	9.5	0.8, 109.2
	Second week of neonate’s life ^e^	Lymphocytosis	*n* = 17 of 28 (61%)	*n* = 45 of 48 (94%)	APC	9.7	2.4, 39.1
			*n* = 3 of 28 (11%)	*n* = 13 of 48 (27%)	KP	3.1	0.8, 12.0
			*n* = 1 of 28 (4%)	*n* = 4 of 48 (8%)	SA	2.5	0.3, 23.1
			*n* = 9 of 28 (32%)	*n* = 24 of 48 (50%)	CoNS	2.1	0.8, 5.6
		CRP > 5 [mg/L]	*n* = 13 of 27 (48%)	*n* = 36 of 38 (95%)	APC	19.4	3.9, 97.1
			*n* = 1 of 27 (4%)	*n* = 7 of 38 (18%)	MS	5.9	0.7, 50.9
			*n* = 3 of 27 (11%)	*n* = 12 of 38 (32%)	EC	3.7	0.9, 14.7
			*n* = 3 of 27 (11%)	*n* = 11 of 38 (29%)	KP	3.3	0.8, 13.1
			*n* = 1 of 27 (4%)	*n* = 4 of 38 (11%)	SA	3.1	0.3, 29.0
			*n* = 8 of 27 (30%)	*n* = 20 of 38 (53%)	CoNS	2.6	0.9, 7.5

^a^ OR calculations included only bacteria significant for the development of ROP (Table 2) and Candida albicans. ^b^ OR > 1 indicates that the occurrence of premature disease/comorbidities in the group with severe ROP or the chance for the occurrence of a given bacteria is OR times higher than in the group without ROP. ^c^ Included diseases *p* < 0.05 and OR ≥ 1.5 for EG. ^d^ Included laboratory parameters *p* < 0.05 and OR ≥ 1.5. Each parameter has been calculated on the basis of mean values from provided norms for the average period GA: CG and EG. ^e^ GA: EG 28 ± 1; CG 30 ± 1. ^f^ Intestinal, inguinal, umbilical required surgery.

## Data Availability

The datasets used and/or analysed during the current study are available from the corresponding author on reasonable request.

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
