# Peer review of "Association of Positive Bacterial Cultures Obtained from the Throat, Anus, Ear, Bronchi and Blood in Very-Low-Birth-Weight Premature Infants with Severe Retinopathy of Prematurity—Own Observations"

_jcm, 2023, doi:10.3390/jcm12196374_

Round 1
Reviewer 1 Report
Dear authors,
The study methodology requires substantial additional descriptions. Describe in section Methods “Microbiological smears and blood“ (more details). In section Microbiology it is mentioned that: ʺIn the analyzed smears there were 25 types of bacteria identified, of which 11 were selected as the most frequent ʺ. The bacteria (Staphylococcus aureus, Klebsiella pneumoniae, Maltophilia Stenotrophomonas, Enterobacter cloacae) were identified on the smear? In Table 3 “Smears from 4 cavities and from blood”, were the bacteria identified on the smear? Also, in Table 3 detailed smears and general smear description is not very clear (m=315, m=255 ???).
We have gone through the manuscript and did find some spelling errors, grammar issues (for example: bac-teria, bron-chi, treat-ed, laborato-ry, etc., and reference format issues that need to be corrected. Abbreviations such as: incl.: e=58 eyes, acc. to, should be corrected. Please define preneonates. Bacterial names should be italicized.
Certificate number of the study KB-0012/131/11/2021/Z has been obtained from Ophthalmology Prematurity Clinic?
Moderate editing of English language required.
Author Response
Author's Reply to the Review Report (Reviewer 1)
The authors of the manuscript thank the reviewers for their detailed review. All comments have been meticulously explained by us and have been added to the text as a supplement. We hope that the current form of the article is still accessible. We are willing to complete the text if you still read it and notice any errors.
Answers for Specific Comments
General Comments:
Dear authors,
The study methodology requires substantial additional descriptions. Describe in section Methods “Microbiological smears and blood“ (more details).
Authors: The authors added the fragment requested by the reviewer in the method section.
In section Microbiology it is mentioned that: ʺIn the analyzed smears there were 25 types of bacteria identified, of which 11 were selected as the most frequent ʺ. The bacteria (Staphylococcus aureus, Klebsiella pneumoniae, Maltophilia Stenotrophomonas, Enterobacter cloacae) were identified on the smear? In Table 3 “Smears from 4 cavities and from blood”, were the bacteria identified on the smear? Also, in Table 3 detailed smears and general smear description is not very clear (m=315, m=255 ???).
Authors: These bacteria were first collected with a swab from 4 body orifices and then identified in the microbiological laboratory by bacterial culture.
Yes, during the analysis of all data collected from the records of premature babies, 25 different species of bacteria were identified: Escherichia coli, Acinetobacter baumanii, Klebsiella pneumoniae, Enterococcus faecialis, Enterococcus faecium, Staphylococcus epidermidis, Klebsiella oxytoca, Staphylococcus haemolyticus, Pseudomonas aeruginosa, Citrobacter freundii, Klebsiella aerogens, Morganiella morgani, Enterobacter cloacae, Serratia marcescens, Staphylococcus aureus, Staphylococcus capitis, Maltophilia stenotrophomonas, Enterobacter asburiae, Staphylococcus hominis, Enterobacter kobei, Ureaplasma, Citrobacter farmerii, Citrobacter braaki, Acinetobacter Iwoffii, Citrobacter murliniae. Some of them appeared at a rare frequency that would not have any statistical significance, so it was decided to exclude all bacteria from the study. We have added all bacteria in the "Microbiology" fragment.
m stands for swab. There were 63 premature babies in the control group, and 51 in the research group. The number of detailed swabs in the research group (m) obtained from 4 body orifices + blood and calculated according to the equation 51x5=255, while in the control group: 63x5=315. Thanks to this, the frequency of occurrence of a specific bacterium in a given premature baby was checked (e.g. if the bacterium appeared both in the anus and in the oral cavity tells us the baby is sicker).
General swabs indicate a generally more frequent occurrence of bacteria in the research group - n means the number of premature babies, i.e. whether a positive bacterial culture was obtained from any of the holes (or from the blood).
We have tried to describe the relationship data below the table for clarification purposes. This is a difficult table, but very important for the article.
We have gone through the manuscript and did find some spelling errors, grammar issues (for example: bac-teria, bron-chi, treat-ed, laborato-ry, etc., and reference format issues that need to be corrected.
Authors: The authors would like to sincerely apologize for the grammatical errors in the manuscript. This was probably due to conversion to a different text format and failure to proofread before submission to the journal. Grammatical errors have been corrected, such as references.
Abbreviations such as: incl.: e=58 eyes, acc. to, should be corrected.
Authors: Abbreviations has been corrected.
Please define preneonates.
Authors: The authors admit that finding a definition of this word may be difficult, so they decided to change the word "preneonates" to "premature babies" throughout the text.
Bacterial names should be italicized.
Authors: Corrected as requested by the reviewer.
Certificate number of the study KB-0012/131/11/2021/Z has been obtained from Ophthalmology Prematurity Clinic?
Authors: The number was obtained after verification by the local bioethics committee.
Thank you for the valuable comments in your review. They helped to change some parts of the article, making them clearer and understandable.
We would like also to thank you for your understanding and the extra time we received.
Best Regards,
Authors
Monika Modrzejewska, Wiktoria Bosy-GÄ…sior & Wilhelm Grzesiak

Reviewer 2 Report
In this study, the authors evaluated the "Bacterial and fungal swabs/infections as a predict factor for retinopathy of prematurity." The study is relevant and will be of great importance to researchers working in this area; however, it was not properly presented. A major revision should be made to improve the manuscript.
Comments are provided below:
The title of the study does not reflect the content of the manuscript; I therefore suggest that the title should be modified to suit the content of the manuscript.
In the abstract, ROP in line 1 should be written in full. The methods, results, and conclusion should be improved.
The introduction is inadequate and should be improved. In the last paragraph of the introduction, this statement, "As there are no publications on the topic in current internet databases such as PubMed, Google Scholar or UpToDate" is not necessary.
In the materials and methods, the criteria for the selection of the neonate should be clearly stated. The methods used for the evaluation of NLR, WBC, NEU, ULN, PLT, CRP, and PRC should be described.
There is no title for Figure 1.
In Table 2, it seems that the SD for NRL, CRP, and PRC is greater than the mean; such results should be presented with a boxplot or better representation, not in tabular form.
In discussion, this statement should be deleted. "Recently published research (library databases such as PubMed, Google Scholar and UpToDate covering years 2011-2021; no publications by Jan 2022) that analyzes prenatal and postnatal infection risk factors shows that they can increase the risk of ROP"
In all the tables, the authors did not present any results for Candida albicans; however, the authors implicated it as the major fungus in preterm infants with severe ROP.
In conclusion, the authors should highlight the major findings of this study, as well as suggest future studies.
Include a list of abbreviations.
English language needs revision (few typographical and grammatical errors were detected).
Extensive editing of English language required
Author Response
Author's Reply to the Review Report (Reviewer 2)
The authors of the manuscript thank the reviewers for their detailed review. All comments have been meticulously explained by us and have been added to the text as a supplement. We hope that the current form of the article is still accessible. We are willing to complete the text if you still read it and notice any errors.
Answers for Specific Comments
General Comments:
In this study, the authors evaluated the "Bacterial and fungal swabs/infections as a predict factor for retinopathy of prematurity." The study is relevant and will be of great importance to researchers working in this area; however, it was not properly presented. A major revision should be made to improve the manuscript.
Authors: The authors would like to thank the reviewer for the positive reaction to the topic presented in the manuscript. We have made every effort to improve it in accordance with the reviewer's valuable comments.
Comments are provided below:
The title of the study does not reflect the content of the manuscript; I therefore suggest that the title should be modified to suit the content of the manuscript.
Authors: We have changed the title to something more related to the topic discussed.
Current title
Association of positive bacterial cultures obtained from the throat, anus and blood in very low birth weight premature infants with severe retinopathy of prematurity - own observations.
In the abstract, ROP in line 1 should be written in full. The methods, results, and conclusion should be improved.
Authors: The shortcut has been expanded. Abstract has been improved.
The introduction is inadequate and should be improved. In the last paragraph of the introduction, this statement, "As there are no publications on the topic in current internet databases such as PubMed, Google Scholar or UpToDate" is not necessary.
Authors: The introduction has been modified according to the reviewer's recommendation. Last paragraph has been removed.
In the materials and methods, the criteria for the selection of the neonate should be clearly stated. The methods used for the evaluation of NLR, WBC, NEU, ULN, PLT, CRP, and PRC should be described.
Authors: Selection of premature neonates had been described in methods section: “Eligibility criteria included: birth weight (BW) ≤1710g and gestational age (GA) ≤32; diagnosis of 3ROP, 5ROP and A-ROP which were treated with diode laser or anti-VEGF intravitreal injections (Ranibizumab) or combined: laser and anti-VEGF.”
The clinical status of ROP was assessed based on the International Classification of ROP, which determined the degree of disease advancement (1ROP - 5ROP and aggressive- ROP (A-ROP), the area of ​​retinal involvement and the presence/absence of "plus" disease; acc. To:
Chiang, M.F.; Quinn, G.E.; Fielder, A.R.; Ostmo, S.R.; Paul Chan, R.V.; Berrocal, A.; Binenbaum, G.; Blair, M.; Peter Campbell, J.; Capone, A.Jr.; et al. International Classification of Retinopathy of Prematurity, Third Edition. Ophthalmology. 2021, 28(10):e51-e68. doi: 10.1016/j.ophtha.2021.05.031.
Inflammatory parameters taken into account by the authors were included because in the literature they are considered additional parameters - predictive of the development of severe ROP. The authors, having this medical data from the information cards, re-analyzed these values ​​and took them into account in their analyses.
We obtained laboratory results from medical information cards - we are ophthalmologists and we focused on examining the relationship between existing parameter records and ROP. It seems to us that deepening the methodology data in the field of medical analysis may introduce chaos in an article that is already quite complicated in content.
There is no title for Figure 1.
Authors: Above Figure 1 there is its title, but we changed it from "The total amount of positive smears from each of the two physiological cavities seperately: throat, anus; and blood per each prematures indicated statistically significant differences between the groups." to: “Graphs indicating the presence of positive bacterial cultures /%/ obtained from the throat, anus and blood cultures in the group of premature babies with severe ROP compared to the group of premature babies without ROP.” Figure 1 also became Figure 2 after changing Table 2 into boxplot.
In Table 2, it seems that the SD for NRL, CRP, and PRC is greater than the mean; such results should be presented with a boxplot or better representation, not in tabular form.
Authors: Following the reviewer's instructions, the authors made a boxplot regarding the parameters mentioned by the reviewer. Table 2 has been moved to additional materials.
Now – Boxplot is Figure 1, old Figure 1 –> Figure 2, Table 3 -> Table 2, Table 4 -> Table 3.
In discussion, this statement should be deleted. "Recently published research (library databases such as PubMed, Google Scholar and UpToDate covering years 2011-2021; no publications by Jan 2022) that analyzes prenatal and postnatal infection risk factors shows that they can increase the risk of ROP"
Authors: This statement was removed by the authors.
In all the tables, the authors did not present any results for Candida albicans; however, the authors implicated it as the major fungus in preterm infants with severe ROP.
Authors: The authors agree with the reviewer - when preparing the table, only bacteria were included because they appeared most often in premature babies. It is true that Candida is the most frequently recognized fungus and appeared significantly more frequently in EG, but we decided to remove the information about Candida from the title because it may be misleading. Candida should be treated as an addition to work, not a significant finding.
In conclusion, the authors should highlight the major findings of this study, as well as suggest future studies.
Authors: The authors summarized the research results indicating that the examination of positive cultures from the pharynx and blood - for KP and Co-negative Staphylococci - are associated with severe ROP which may have predictive significance in determining the future earlier starting of treatment of ROP
Include a list of abbreviations.
Authors: List of abbreviations included on the beginning of the manuscript. After creating the list of abbreviations at the beginning of the manuscript, the abbreviations below the tables were removed.
English language needs revision (few typographical and grammatical errors were detected).
Authors: Grammatical errors corrected.
Thank you for the valuable comments in your review. They helped to change some parts of the article, making them clearer and understandable.
We would like also to thank you for your understanding and the extra time we received.
Best Regards,
Authors
Monika Modrzejewska, Wiktoria Bosy-GÄ…sior & Wilhelm Grzesiak

Round 2
Reviewer 1 Report
In abstract, it is necessary to change the secretion collection technique. In your answer you mentioned: " These bacteria were first collected with a swab from 4 body orifices and then identified in the microbiological laboratory by bacterial culture." Please make changes in the article (the samples were not worked on smears, they are CULTURES !!!)
In section "Microbiological smears and bloodʺ Please provide more details regarding incubate temperature, time, etc.
Minor editing of English language required
Author Response
Author's Reply to the Review Report (Reviewer 1)
The authors of the manuscript thank the reviewers for their detailed review. All comments have been meticulously explained by us and have been added to the text as a supplement. We hope that the current form of the article is still accessible. We are willing to complete the text if you still read it and notice any errors.
In abstract, it is necessary to change the secretion collection technique. In your answer you mentioned: " These bacteria were first collected with a swab from 4 body orifices and then identified in the microbiological laboratory by bacterial culture." Please make changes in the article (the samples were not worked on smears, they are CULTURES !!!)
Authors: As requested by the reviewer, the word "smear" has been replaced with "cultures" throughout the text. We fully agree with the reviewer's comment and thank you for this advice.
In section "Microbiological smears and bloodʺ Please provide more details regarding incubate temperature, time, etc.
Authors: At the reviewer's request, we performed an analysis in the microbiology laboratory and learned about the methods of cultivating microorganisms. We have modified this part in the methods.
Minor editing of English language required
Authors: The English language in the manuscript has been improved.
Thank you for the valuable comments in your review. They helped to change some parts of the article, making them clearer and understandable.
Best Regards,
Authors
Monika Modrzejewska, Wiktoria Bosy-GÄ…sior & Wilhelm Grzesiak
